# Two-photon fluorescence imaging and specifically biosensing of norepinephrine on a 100-ms timescale

Leiwen Mao[1], Yujie Han[1], Qi-Wei Zhang [1] ✉ & Yang Tian [1] ✉

Norepinephrine (NE) is a key neurotransmitter in the central nervous system of organisms; however, specifically tracking the transient NE dynamics with high spatiotemporal resolution in living systems remains a great challenge. Herein, we develop a small molecular fluorescent probe that can precisely anchor on neuronal cytomembranes and specifically respond to NE on a 100-ms timescale. A unique dual acceleration mechanism of molecular-folding and water-bridging is disclosed, which boosts the reaction kinetics by $>10^5$ and $>10^3$ times, respectively. Benefiting from its excellent spatiotemporal resolution, the probe is applied to monitor NE dynamics at the single-neuron level, thereby, successfully snapshotting the fast fluctuation of NE levels at neuronal cytomembranes within 2 s. Moreover, two-photon fluorescence imaging of acute brain tissue slices reveals a close correlation between downregulated NE levels and Alzheimer's disease pathology as well as antioxidant therapy.

Neurotransmitters play a crucial role in maintaining physiological processes, such as regulating metabolism, participating in neuromodulation and influencing organ function[1–4]. In particular, signaling between neurons occurs when neurotransmitters are released from the presynaptic membrane, diffuse across the synapse to neighboring neurons, and bind to receptors on the postsynaptic neuronal membrane[5]. For example, dopamine (DA), epinephrine (EP), and norepinephrine (NE) are a class of catecholamine-based structurally similar and interconvertible neurotransmitters, among which NE is at the central position during their interconversion processes[6,7]. Therefore, NE is considered as one of the key neurotransmitters in the central nervous system of vertebrate organisms, and the dysfunction of noradrenergic transmission is closely related to a series of neurodegenerative and psychiatric disorders, including Parkinson's disease (PD), Alzheimer's disease (AD), depression, etc[8–10]. Despite its well-recognized importance in a variety of physiological and pathophysiological processes, monitoring the transient NE dynamics in living systems with high sensitivity and high spatiotemporal resolution remains a formidable challenge.

In this context, fluorescent probes combined with fluorescence imaging technology may serve as a promising candidate to monitor

biomolecules, including neurotransmitters, in real-time with high sensitivity and spatiotemporal resolution[11–18]. For example, genetically encoded fluorescent protein (GEFP)-based sensors have recently gained much attention for NE sensing and imaging, owing to their cell-specific spatial resolution and rapid kinetics[19,20]. However, the poor ability in distinguishing NE from EP, which differ from each other by only a terminal methyl group, as well as the difficulty of precise anchoring to organelles of interest, have limited their practical application. In contrast, organic small molecular probes are expected to achieve higher specificity for NE detection via tailoring the recognition sites of the probes[21]. Yin's group has developed a pioneering NE-specific probe employing a protect-deprotect strategy-based nucleophilic substitution reaction[22,23]. Nonetheless, the sensing performance of the probe still need to be improved, since interference from DA and EP was around 20%. In addition, it takes tens of minutes to complete the reaction. To the best of our knowledge, there have been no reports on small molecular NE probes with sub-second reaction kinetics, which is essential for monitoring the transient NE dynamics in living systems. What's more, presently reported probes also lack sufficient spatial resolution to label the neuronal cytomembrane, which is the frontier organelle at synapses that is directly involved in neurotransmitter

[1]Shanghai Key Laboratory of Green Chemistry and Chemical Processes, Department of Chemistry, School of Chemistry and Molecular Engineering, East China Normal University, Shanghai, China. ✉e-mail: qwzhang@chem.ecnu.edu.cn; ytian@chem.ecnu.edu.cn

secretion and binding, playing an extremely important role in signaling[24,25]. Therefore, there is an urgent need to develop powerful molecular fluorescent probes to achieve high specificity, high sensitivity, and in particular, high spatiotemporal resolution for direct visualization of transient NE dynamics in living neurons, which may help to deepen the understanding of the physiological and pathological functions of NE at the single-cell level[26].

Herein, we designed and synthesized a novel small molecular fluorescent probe (BPS3, Fig. 1), bearing two phenyl pyridiniums linked by an alkyl chain, which not only guaranteed good water solubility and a flexible molecular conformation, but also exhibited excellent cytomembrane targeting. In addition, the introduction of a terminal *S-p*-toluene carbonothioate as the triggering group for NE, which could undergo a sequential nucleophilic substitution followed by a cyclization reaction to the characteristic primary amino group and β-hydroxyl group of NE, respectively, achieved excellent selectivity toward NE over the other two catecholamine neurotransmitters (EP and AD). The most noteworthy feature of the developed probe BPS3 is its ultrafast reaction kinetics, enabling NE detection within an impressively short time down to 100 ms, which is the most rapid small molecular probe reported by far for quantitative NE sensing and imaging. Through the combination of experimental and theoretical analyses, we revealed a unique dual acceleration mechanism, namely, molecular

conformational folding and water bridging, which significantly reduced the activation energy of the reaction and increased the reaction rate by more than $10^5$ and $10^3$ times, respectively. Finally, benefiting from the superior properties including high specificity, nanomolar sensitivity, neuronal cytomembrane-labeling and 100-ms temporal resolution, as well as near infrared (NIR) two-photon excitation, we applied probe BPS3 to real-time imaging of NE fluctuations in living neurons under external chemical stimulation. The rapid and significant fluorescence variations at the neuronal cytomembrane clearly indicated the transient neuronal NE cytosolic process (within 2 s) upon high potassium stimulation, validating the high spatiotemporal resolution of the probe, which was very beneficial for in-depth studies of the transient NE dynamics and functions during physiological processes in living neurons. Moreover, by two-photon fluorescence imaging of acute mouse brain tissue slices, we revealed the close correlations between downregulated NE levels and AD pathology as well as antioxidant therapy.

## Results
### Fluorescence sensing of NE and the reaction mechanisms
The synthesis of the probe BPS3 was detailed in the Supplementary Information, and its response to NE was initially investigated by UV−vis absorption and fluorescence emission spectroscopy in

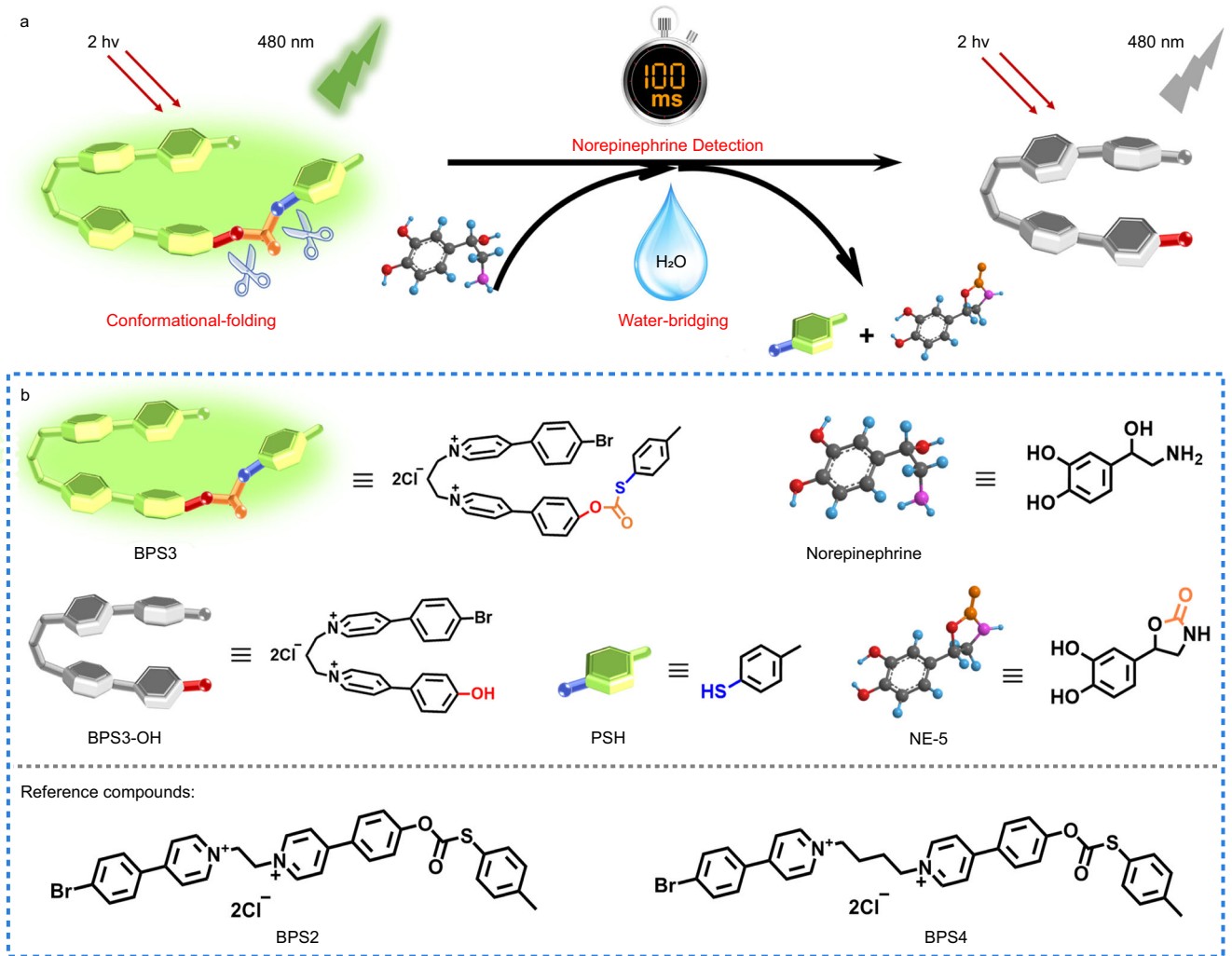

**Fig. 1 | Schematic of the sensing mechanism and the chemical structures of relevant compounds. a** Schematic illustration of the ultrafast sensing mechanism toward norepinephrine by the two-photon fluorescence probe BPS3, boosted by the dual acceleration of conformational-folding and water-bridging. **b** Molecular structures of the compounds referred in this work.

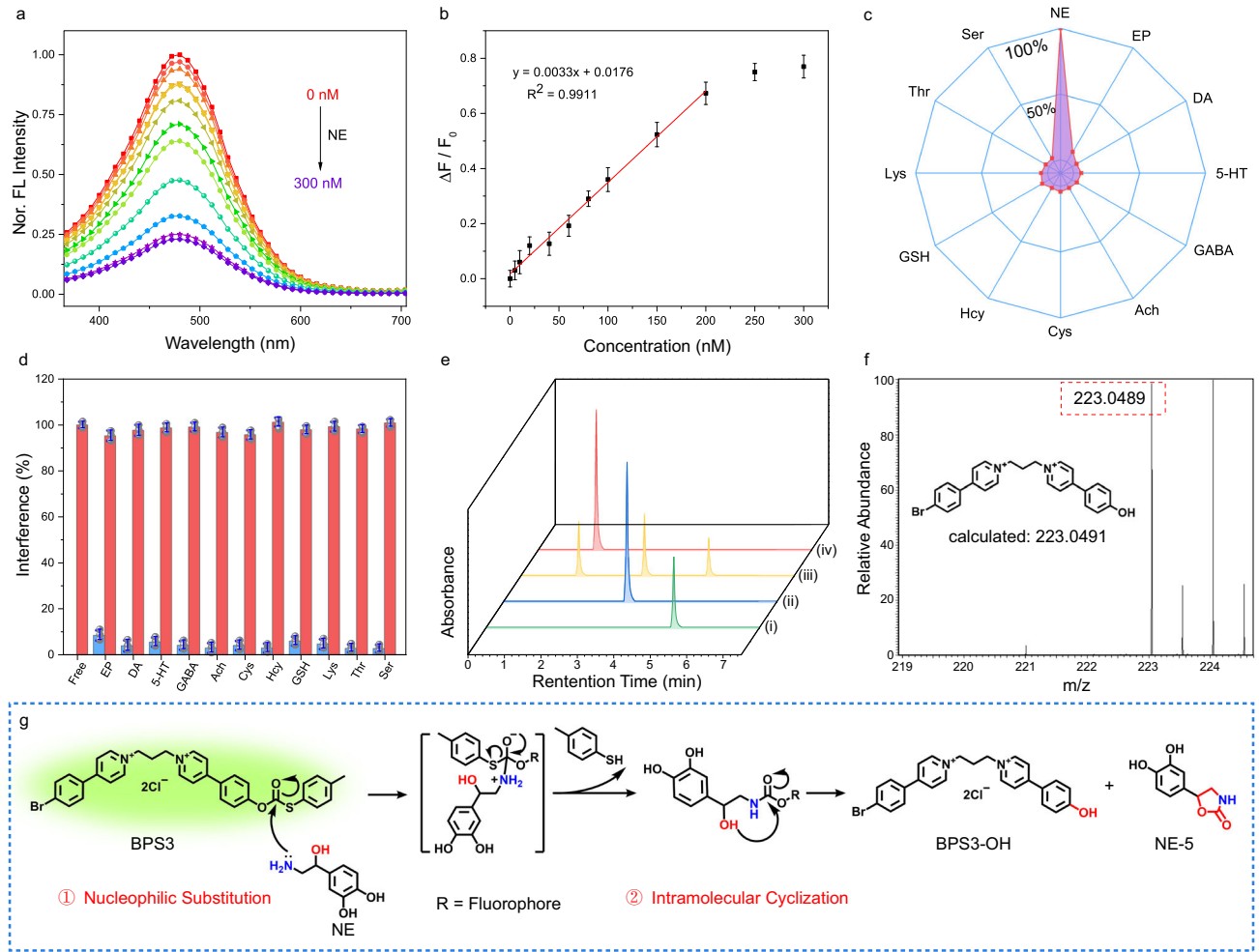

**Fig. 2 | Sensing performances and mechanism. a** Fluorescence spectrum of BPS3 (5 μM) upon addition of NE (0–300 nM), two-photon excited at 720 nm. **b** Plot and linear fitting of the fluorescence variation rates (ΔF/F₀) at 480 nm versus the concentration of NE (0–300 nM). Data are presented as mean ± S.D. Error bars: S.D., $n = 3$ independent experiments. **c** Selectivity and (**d**) competition tests of the probe BPS3 toward NE against other neuron transmitters and amino acids (norepinephrine (NE), epinephrine (EP), dopamine (DA), serotonin (5-HT), γ-aminobutyric acid (GABA), acetylcholine (Ach), cysteine (Cys), homocysteine (Hcy), glutathione (GSH), lysine (Lys), threonine (Thr), serine (Ser)), measured after 10 min of mixing. The concentrations of NE, EP, and DA were 300 nM, and other species were all 1 mM. Data are presented as mean ± S.D. Error bars: S.D., $n = 3$ independent experiments. **e** HPLC spectra of (i) NE, (ii) BPS3, (iii) BPS3 reacted with NE, and (iv) BPS3-OH. **f** HR-MS spectrum of the product of BPS3 reacted with NE in aqueous solution. **g** The sequential nucleophilic substitution-cyclization mechanism of the reaction between probe BPS3 and NE.

phosphate-buffered saline (PBS, 10 mM, pH = 7.4). As shown in Supplementary Fig. 35, the probe had a main absorption peak at 360 nm, whose intensity was gradually decreased upon addition of NE (0−300 nM). Accordingly, the fluorescence emission at 480 nm was also significantly quenched after incubation with NE (Supplementary Fig. 36). Considering the potential for deeper penetration and lower photodamage to living systems in biological applications, the NIR two-photon absorption (TPA) performance of the probe BPS3 was then assessed. As shown in Supplementary Fig. 37, BPS3 showed satisfactory two-photon absorption in water with a maximum TPA cross-section value of 44.6 GM at 720 nm (Rhodamine B as a standard, in the optimized solvent: ethanol). Therefore, 720 nm was chosen as the optimal two-photon excitation wavelength for the following experiments.

Subsequently, the fluorescence titration experiment was carried out in buffer solution for the probe BPS3 upon addition of NE. As shown in Fig. 2a, with the gradual addition of NE, the fluorescence intensity of the probe at 480 nm decreased accordingly, which almost reached the minimum (ca. 23% of the initial value) when 300 nM of NE was added. Fig. 2b plotted the fluorescence intensity variation rate (ΔF/F₀) versus the concentration of NE, which exhibited good linearity

within the concentration range of 0−200 nM, and the detection limit was evaluated to be 0.5 nM (3σ/S, where σ is the standard deviation (S.D.) of the probe sample, $n = 20$, and S is the slope of calibration curve). These results indicated that the developed probe BPS3 could respond to NE with ultra-high sensitivity, which met the detection requirement of low-concentration of NE in live neurons (1–100 nM)[27]. Furthermore, the selectivity of the probe BPS3 toward NE was then investigated comprehensively by incubation of the probe with various biologically related analytes in PBS buffer, such as potential competing neurotransmitters, amino acids, metal ions, as well as reactive oxygen species (ROS). As shown in Supplementary Fig. 38, except for NE, the addition of other related neurotransmitters hardly affected the fluorescence emission of the probe. Notably, even for the catecholamine neurotransmitters that were generally indistinguishable from NE, i.e., DA and EP, their interference was negligible (4.1% for DA, and 8.6% for EP as compared to that of the target analyte NE, Fig. 2c), verifying the superb selectivity of BPS3 toward NE. On the other hand, the fluorescence variations of BPS3 mixed with NE were hardly affected in the presence of amino acids, metal ions and other related biological interferents, indicating the good anti-interference ability of the probe (Supplementary Fig. 39). What's more, the impact of pH variations and

photo irradiations on the fluorescence emission of the probe BPS3 and the mixture BPS3 + NE were also assessed. The results showed that both solutions exhibited stable emissions at different pH values ranging from 5.0 to 9.0 (Supplementary Fig. 40), or irradiated by a Xe lamp (90 W) for 2 h (Supplementary Fig. 41), indicating their good pH and photo stability. Therefore, these in vitro experiments have clearly verified the potential of BPS3 to monitor NE under complicated physiological conditions with high sensitivity, selectivity, as well as stability.

To further understand the reaction procedure, high-performance liquid chromatography (HPLC) was then used to track the reaction between probe BPS3 and NE. As shown in Fig. 2e, the reaction mixture of BPS3 incubated with NE (curve iii) revealed three substances with well distinguished retention times of 5.2, 3.4, and 1.6 min, which could be assigned to the residual NE, probe BPS3, and the cleaved product BPS3-OH, respectively, by comparison with the corresponding reference samples (curves i, ii, and iv). Moreover, the high-resolution ESI mass spectrometry (HR-MS) analysis of the mixed BPS3 and NE aqueous solution again confirmed the existence of the cleaved product BPS3-OH (m/z = 223.0489, calcd for $[C_{25}H_{23}BrN_2O]^{2+}/2$, $([BPS3-OH-2Cl]^{2+}/2)$, Fig. 2f), and a cyclized NE-derivative NE-5 (m/z = 194.0459, calcd for $[C_9H_8NO_4]^-$, $([NE-5-H]^-)$, Supplementary Fig. 42). The final evidence of the reaction mechanism was provided by $^1H$ NMR spectra for the in-situ reaction of BPS3 with NE in deuterated water, in which all the cleavage products were observed and clearly assigned as BPS3-OH, NE-5, and the free p-toluenethiol compound (PSH), respectively (Supplementary Fig. 43). Therefore, based on the above results, the reaction process between the probe BPS3 and NE was confirmed, which involved a sequential nucleophilic substitution-cyclization reaction to form an uncaged product BPS3-OH, as schemed in Fig. 2g. This delicate reaction process contributed to the excellent selectivity of NE over EP and DA, because the increased steric hindrance effect of the secondary amine in EP greatly hindered the reaction efficiency, while due to the lack of β-OH, DA was almost unreactive with the probe within the measured time (Supplementary Fig. 44).

After demonstrating the reaction procedure, the fluorescence response mechanism of BPS3 toward NE was subsequently studied. First, two reference compounds, namely R1, R2 (Supplementary Fig. 45a), which represented the two fragment moieties of the probe BPS3, respectively. As shown in Supplementary Fig. 45b, the p-bromophenyl pyridinium moiety (R1) emitted little fluorescence, while the S-phenyl carbonothioate-containing pyridinium moiety (R2) exhibited similar fluorescence emission to the probe BPS3. Therefore, it could be inferred that the fluorescence of the probe was generated from the S-phenyl carbonothioate-containing pyridinium moiety. Next, in order to rationalize the fluorescence response caused by the cleavage reaction triggered by NE, we further synthesized the third reference compound (R3) representing the fragment moiety of the product BPS3-OH (Supplementary Fig. 45a). It could be seen from Supplementary Fig. 45b that after cleavage of the S-phenyl carbonothioate group, and form the hydroxyl product R3, the fluorescence was significantly quenched as compared with R2. We deduced that in both the reference compound R3 and product BPS3-OH, the hydroxyl group acted as an electron-donating group and the pyridinium acted as the electron-withdrawing group, thus generating a twisted intramolecular charge transfer (TICT) state, which greatly enhanced the non-radiative decay thus quenched the fluorescence emission. To verify this inference, the viscosity-dependent absorption spectra were measured for BPS3-OH. As shown in Supplementary Fig. 46a, as the viscosity of the solvent increased by increasing the fraction of glycerol in solution, the absorption maximum underwent an obvious red-shifting from 354 nm to 362 nm. Such viscosity-dependent solvatochromism clearly confirmed the TICT feature of the compound BPS3-OH. Moreover, to get a deep insight into the photophysical property of BPS3-OH, the femtosecond transient absorption spectrum was measured in PBS (10 mM,

pH = 7.4) with the probe delay time from 0.001 to 200 ps. As shown in Supplementary Fig. 46b, the initial evolution spectrum of BPS3-OH from 0.1 to 2.2 ps showed a significant negative signal from 400 to 600 nm, which was consistent with fluorescence spectrum and could be assigned to the stimulated emission (SE) of locally excited (LE) state. Next, this negative signal underwent decay accompanied by the appearance of a positive signal from 590 to 720 nm which was attributed to the absorption of LE state. At about 4.6 ps, another absorption signal at 460 nm appeared which slightly blue-shifted with time evolution, and vanished at around 30 ps. This signal was assigned to the absorption of TICT state[28]. Therefore, all these results have confirmed the TICT characteristics of the NE-triggered cleaved product BPS3-OH. On the contrary, the TICT process did not occur in the probe BPS3 due to the existence of electron-withdrawing groups on both sides of the fluorophore. This may be the reason for the significantly quenched fluorescence emission of the probe after reaction with NE.

## Mechanisms of the conformation-dependent reaction kinetics for NE sensing

Next, attention was paid to the study of the response rate of the probe BPS3 toward NE, by using a rapid-mixing stopped-flow technique, which had a finite mixing time of less than 8 ms[29]. Fig. 3a showed the kinetics data obtained for 5 μM of probe BPS3 reacted with 100 nM of NE in aqueous solution, in which the fluorescence intensity of the probe at 480 nm decreased immediately after mixing with NE, and reached a plateau within 100 ms (the green curve), with an observed reaction rate constant ($k_{obs}$) as fast as 53.2 $s^{-1}$ (Supplementary Fig. 47)[30]. The concentration-dependent kinetics studies revealed that different concentrations of NE required different time to equilibrate the reaction, where the lower the concentration, the shorter the time required (Supplementary Fig. 48). Considering the physiological concentration of NE in neurons (1–100 nM), the developed probe BPS3 could detect NE within 100 ms. Such a fast reaction rate has not been reported on other small molecular NE probes so far, whose detection time is generally on the timescale of minutes or even tens of minutes. This exciting finding strongly motivated us to explore the reaction mechanism in depth.

Subsequent study was first focused on the structural analysis of the probe itself. As can be seen in Fig. 1, the structure of BPS3 consists an alkyl chain linking two aromatic moieties, which may lead to a flexible conformation. Our previous studies have confirmed that such molecules may exist in folded or stretched conformations, depending on the surrounding environment, which could be verified by examining the interproton nuclear Overhauser effect (NOE) in the molecule using 2D NOESY or ROESY NMR technologies[31–33]. As shown in Fig. 3b, couples of NOE signals were found between protons $H_{1,2}$ from the boromophenyl group and protons $H_{11,13}$ from the carbonothioate phenyl moieties, confirming the steric proximity of the two terminal groups, which is a typical characteristic of folded conformation. In addition, theoretical calculations for different conformations of the probe BPS3 also confirmed that the folded conformation had relatively lower energy than the stretched or quasi-orthogonal conformations (Supplementary Fig. 49). The higher stability of the folded conformation could be further rationalized by the independent gradient model based on Hirshfeld partition (IGMH) analysis (Fig. 3c)[34], which showed obvious van der Waals interactions and π-π stacking effects between the benzene rings (green lamellar regions) that were not found in the stretched structure. Furthermore, compared with the stretched conformation, the carbonothioate carbon of BPS3 has a larger Hirshfeld Charge value in the folded conformation (Fig. 3d), indicating its stronger electrophilicity, which is more favorable for the nucleophilic substitution reaction[35].

To see if it was indeed the case that the folded conformation of the probe could facilitate the reaction with NE, we designed and synthesized two control compounds that were structurally analogous to

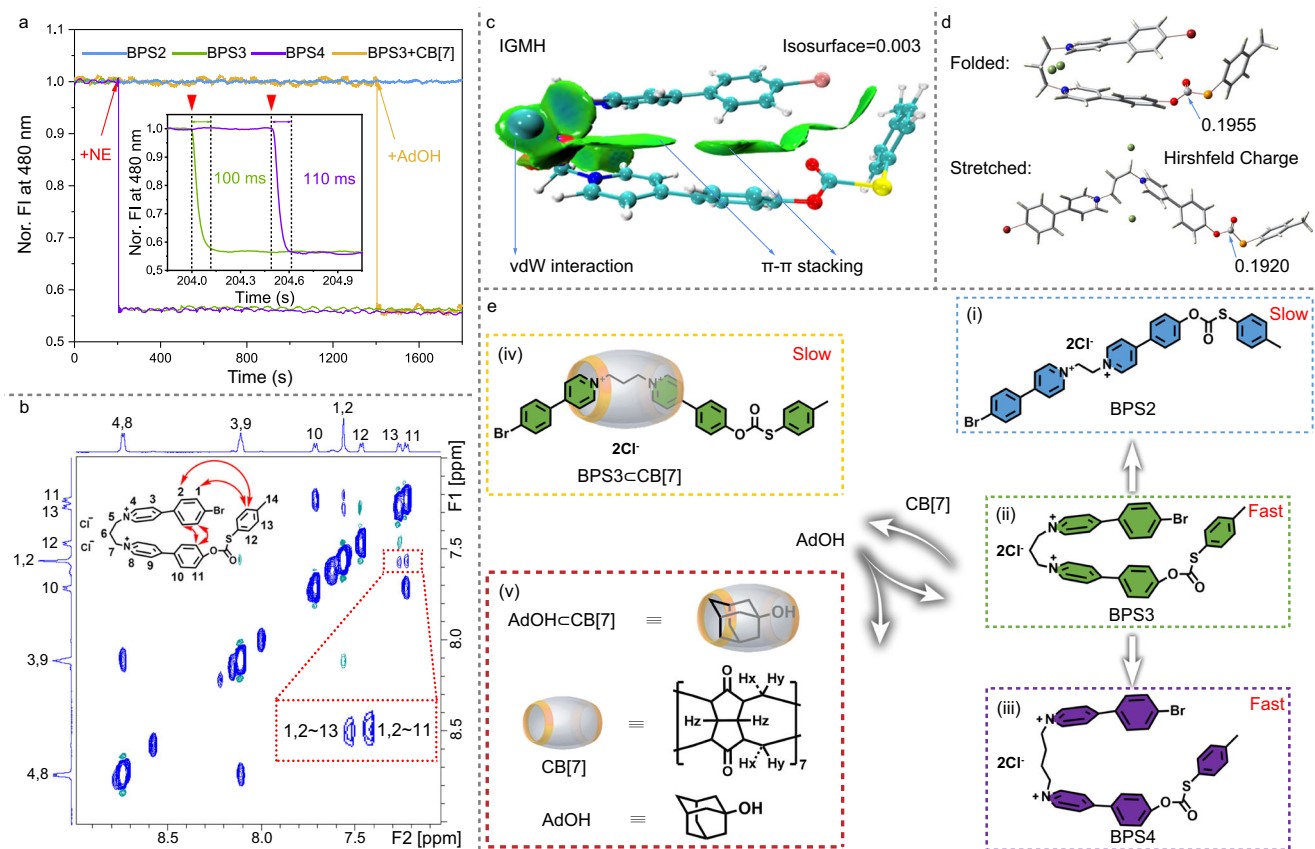

**Fig. 3 | Conformation-dependent reaction kinetics for NE sensing. a** Normalized fluorescence response dynamics (recorded at 480 nm, excited at 720 nm) of 5 μM aqueous solution of BPS2, BPS3, BPS4, and BPS3 + CB[7] (1 equiv.) with addition of 100 nM NE. For the curve BPS3 + CB[7], an additional 1 equiv. of AdOH was added at the time of 1200 s after the addition of NE. Inset: Enlarged view of the partial curve for the response of BPS3 and BPS4 toward NE. **b** Partial 2D NOESY NMR spectrum of BPS3 (1 mM in D₂O), at 298 K. **c** IGMH isosurface of BPS3 in folded conformation. **d** The calculated Hirshfeld Charge of the carbonothioate carbon of BPS3 in folded and stretched conformations, respectively. **e** Schematic of molecular conformational regulations by covalent or supramolecular approaches.

the probe BPS3, named BPS2 and BPS4, respectively, which had identical functional groups and reactive sites, but only differed in the length of the alkyl chain (Fig. 1b bottom, and Fig. 3e, i–iii). Obviously, since there were only two carbons in the alkyl chain of the BPS2 molecule, it could hardly form a folded conformation due to the larger molecular bond tension and steric hindrance; while compound BPS4, with a four-carbon alkyl chain, should have sufficient molecular flexibility to form a folded conformation as the probe BPS3 do. Whereafter, the reaction kinetics of these two control compounds with NE were subsequently studied and compared with BPS3. It was quite surprising that despite with only one-carbon shorter in the alkyl chain, compound BPS2 hardly produced any response to NE at the same condition (blue curve in Fig. 3a), even extending the reaction time to 7 days (Supplementary Fig. 50a). A very slow reaction could be observed when the reaction temperature was gradually increased, e.g., the reaction could not be finished in 12 h at 50 °C (Supplementary Fig. 50b); while it took nearly 8 h at 60 °C, with a $k_{obs}$ value of $1.6 \times 10^{-4}\,s^{-1}$ (Supplementary Fig. 50c, e); and even at 70 °C, it still required ca. 60 min to complete the reaction ($k_{obs} = 1.25 \times 10^{-3}\,s^{-1}$, Supplementary Fig. 50d, f). In contrast, the compound BPS4, with a four-carbon alkyl chain in the structure, exhibited almost comparable kinetics as the probe BPS3 and could complete the reaction to NE within 110 ms at room temperature with a $k_{obs}$ value of 35.5 s⁻¹ (purple curve in Fig. 3a, and Supplementary Fig. 47). Therefore, the observed reaction rate constants for compounds BPS3 and BPS4 (with folded conformations) at room temperature are more than $10^5$ times larger than that of compound BPS2 (with a stretched conformation) even at 60 °C. Besides, we also synthesized the control compound R2 as the

half-probe and studied its reaction kinetics toward NE, and compared it with the probe BPS3. As shown in Supplementary Fig. 51, although it had the same reaction site as BPS3, the reaction rate of the control compound R2 toward NE was much lower than that of the probe BPS3. Under the same conditions, it took about 257 seconds to reach the reaction equilibrium, while only 100 ms for the probe BPS3 (Fig. 3a), a difference of ca. 2570 times.

Next, we wondered whether there was a way to directly manipulate the conformation of BPS3 itself from its original folded conformation to a stretched one, without changing its covalent backbone, and to see if it could modulate its reaction kinetics with NE. To this end, a delicate supramolecular approach was designed (Fig. 3e, iv, v). Considering its strong binding affinity to alkyl dipyridinium derivatives as well as the rigid cavity, the macrocycle cucurbit[7]uril (CB[7])[36–38] was introduced as a host molecule to bind with BPS3. As shown in Supplementary Fig. 52, the host-guest structure was studied by 2D ROESY NMR spectroscopy, in which strong NOE signals were found between protons $H_{3,4,6,8,9}$ of BPS3 and the inward proton $H_x$ of CB[7]. These cross peaks clearly confirmed the threading of CB[7] onto the guest molecule BPS3, and located at the alkyl dipyridinium site. Benefiting from the rigid cavity of CB[7] and the precisely localized self-assembly, which prevented the folding of the guest molecule, we successfully obtained a stretched conformation of BPS3 without covering the reactive site for nucleophilic attack (the carbonothioate group) (Fig. 3e, iv). Subsequently, we studied the reaction kinetics of this supramolecular probe system in which BPS3 was fixed in the stretched conformation. The result was as expected, i.e., when NE was added, the probe solution hardly had any fluorescence response even

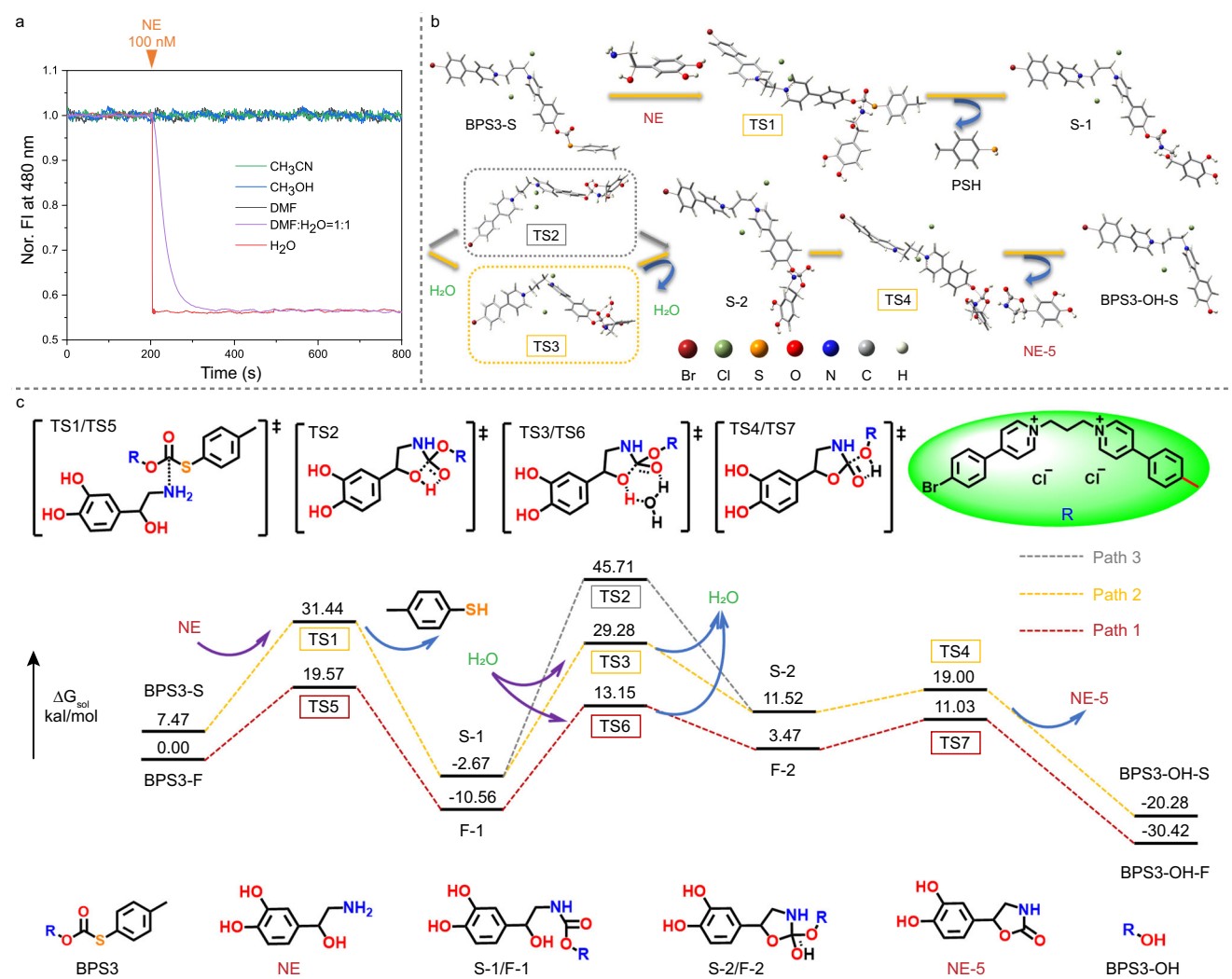

**Fig. 4 | Mechanism studies of the water-bridging-boosted reaction kinetics for NE detection. a** Normalized fluorescence response dynamics (recorded at 480 nm, excited at 720 nm) of 5 µM BPS3 solution in different solvents with addition of 100 nM NE. **b** The pathway and possible transition states of the reactions between the stretched probe BPS3 and NE, with or without the involvement of $H_2O$. **c** Calculated relative energies of the compounds and possible transition states during the reactions between the probe BPS3 and NE in different pathways.

after waiting for more than 20 min (yellow curve in Fig. 3a). In contrast, if 1-adamantanol (AdOH, a stronger binding guest molecule for CB[7])[39,40] was subsequently added to the above solution, the fluorescence was quenched immediately, indicating the regain of the ultrafast reaction kinetics (yellow curve in Fig. 3a). This was due to the stronger competitive binding of AdOH with CB[7], which led to the dissociation of BPS3 from the cavity of CB[7] and recovering its folded conformation in solution (Fig. 3e, v). Therefore, through the above series of conformational regulation and kinetics experiments, the crucial role of the folded conformation of the probe in accelerating its reaction rate with NE was undoubtedly confirmed.

## Mechanisms of the water-bridging-boosted reaction kinetics in NE detection

After recognizing the important influence of molecular conformation, we further explored the effect of reaction transition states on reaction kinetics, which was inspired by our unexpected finding of the water-dependent reaction rate. As noted above, the probe BPS3 could complete the detection of NE within 100 ms in aqueous solution. However, if the water fraction in the system was reduced to 50% by mixing with dimethylformamide (DMF), it would take ca. 120 s to complete the reaction with NE (Fig. 4a, purple curve), with a $k_{obs}$ value

of $4.36 \times 10^{-2}\,s^{-1}$ (Supplementary Fig. 53), which was about 1200 times slower as compared with that in the pure water system (53.2 $s^{-1}$). Moreover, if the probe was mixed with NE in pure organic solvents (DMF, $CH_3OH$, $CH_3CN$), no reaction could be detected (Fig. 4a). Therefore, it is inferred that the water solvent played an important role in the reaction process between BPS3 and NE.

To gain insight into the mechanism of this water-promoted reaction kinetics, the structures and relative energies of the corresponding intermediates and transition states were calculated and optimized at the PBE0(D3BJ)/6-31 G(d)/SDD level with the SMD solvation model[41–44] in water for both the folded and stretched form of BPS3, denoted as BPS3-F and BPS3-S, respectively. For brevity, we referred to the BPS3-F-involved reaction pathway in water as Path 1, and the BPS3-S-involved one as Path 2. As shown in (Fig. 4b and Supplementary Fig. 54), the theoretical studies of the transition states suggested a gradual and cascade nucleophilic substitution-cyclization pathway. In both pathways (Path 1 and Path 2), a reaction coordinate involving anisotropic H-bonding between the BPS3, the NE fragments and a water molecule was observed in aqueous environment, in which an active water molecule served as a molecular bridge for proton transfer via the formation of an intermolecular six-membered ring transition state (state TS6 in Path 1 and state TS3 in Path 2, in Fig. 4c). In contrast,

calculations of the reaction pathway in a non-aqueous system (Path 3) showed that the cyclization and proton transfer processes of the conjugate could only be achieved through an intramolecular four-membered ring transition state (state TS2, marked in a gray box in Fig. 4b, c), which needed to overcome a much higher reaction energy barrier (48.38 kcal/mol) than that of the intermolecular six-membered ring in Path 2 (31.95 kcal/mol) and Path 1 (23.71 kcal/mol). Therefore, theoretical studies have confirmed that the participation of a water molecule can effectively reduce the activation energy of the cyclization process which is the rate-determining step, by promoting the proton transfer through an intermolecular donor-acceptor molecular bridge. Such a water-bridging-boosted pathway, which is different from the generally believed mechanism of four-membered ring-based direct intramolecular cyclization, can well rationalize the experimental results of the dramatically shortened reaction time in water.

So far, based on the analysis of experiments and theoretical calculations, we have successfully revealed the regulatory factors of the kinetics on the reaction between the probe BPS3 and NE, and revealed a new dual acceleration mechanism of molecular conformational-folding and water-bridging, which enabled the probe with ultrafast response toward NE on a 100-ms timescale, satisfying the temporal resolution for monitoring the NE dynamics in living systems.

### Two-photon fluorescence imaging of NE of living neurons and acute brain slices

Encouraged by the above in vitro results, in this section, the performance of probe BPS3 for detection and imaging of NE in living systems was examined. First, the cytotoxicity and biocompatibility of BPS3 were evaluated by means of a standard MTT assay and fluorescence activated cell sorting (FACS) analysis[45]. As shown in Supplementary Fig. 55, when treated with BPS3 at different concentrations (0, 5, 10, 15, 20, 25, and 30 μM, respectively) for 24 h, the viability of neurons remained around 90%. Meanwhile, the FACS results further confirmed that there was no significant difference between the BPS3-treated (10, 20, and 30 μM) neurons and the control (Supplementary Fig. 56). Thus, both experiments verified the low cytotoxic and good biocompatibility of probe BPS3. Moreover, a colocalization experiment for neurons was further performed by co-staining neurons with probe BPS3 and a commercial membrane dye DiI for 30 min[46]. As shown in Fig. 5a, fluorescence distributions of the two emission channels from BPS3 and DiI overlapped well, with a high Pearson correlation coefficient value of 0.93, indicating its excellent ability in neuronal cytomembrane targeting. It is inferred that the cationic pyridinium have electrostatic affinity with the phosphate anion of the cell membrane surface, while the hydrophobic phenyl groups can be embedded into the cell membrane, contributing to the cell membrane targeting ability of the probe BPS3 (Supplementary Fig. 57)[47,48]. It is worth mentioning that this neuronal cytomembrane-anchored probe is highly beneficial to monitor the fusion process of neurotransmitters at synapses with high spatial resolution at the single-cell level.

On the basis of these experiments, we then intended to monitor the secretion process of endogenous NE in living neurons to reveal the corresponding neuronal response and its dynamics under external stimuli. Here, high concentration of potassium ion was used as a mimic of electrical impulse to generate the action potential, which may further induce neuronal NE response[49]. Experimentally, neurons from the same living mouse brain were first divided into two groups and incubated with probe BPS3 for 30 min. As shown in the Fig. 5b, initially, both the two groups of neurons exhibited strong and comparative fluorescence emission at the neuron cytomembranes. However, when one set was treated with PBS buffer only, while the other with high concentration of potassium, the fluorescence intensities of the two groups differed greatly, that is, the fluorescence intensity of the former almost kept constant, while that of the latter varied significantly within 2 s, indicating a rapid increase of NE at the neuronal cytomembrane in

this group (Fig. 5c). This may be due to the depolarization and opening of the voltage-gated calcium channels on the cell membrane under the stimulation of high concentration potassium, which results in the influx of calcium driven by the electrochemical potential. The increased intracellular calcium concentration further leads to the migration and fusion of NE-containing vesicles to the cell membrane, where they are eventually exocytosed into the synaptic cleft (Fig. 5d)[50]. Therefore, probe BPS3 enabled us for the first time to successfully monitor and snapshot the fast NE response of neurons with precise cytomembrane-targeting and seconds-level spatiotemporal resolution by a small molecular fluorescence probe.

In addition, to gain insight into the correlation between NE and neurodegenerative diseases, such as AD, probe BPS3 was further used to detect and compare the NE levels in AD and normal mouse brains. First, as a two-photon fluorescence probe, the imaging capacity of BPS3 in brain tissue slices was explored (Fig. 5e), and the result showed that satisfactory fluorescent signals could be observed even at a depth of 300 μm, demonstrating its potential in-depth imaging. We then prepared acute brain tissue slices from four different regions of AD and normal mouse, namely, cornu ammonis of hippocampus (CA1), primary somatosensory cortex (S1BF), laterodorsal thalamic nucleus (LD), and caudate putamen (CPu), and then cultured with BPS3[51]. Confocal microscopic images of these tissue slices were presented in Fig. 5f, where the pseudocolor ranging from blue to red represented increase of the relative fluorescence intensity. Then, by randomly selecting 25 cells in each image, followed by statistical analysis on their fluorescence intensity variations, the histogram was generated as Fig. 5g. The statistical results indicated some inhomogeneity of NE distribution in various brain regions. It is worth noting that under the AD model, the NE content in all these brain regions decreased significantly to 58.3%, 48.0%, 58.0%, and 63.9% of the normal level, in CA1, S1BF, LD, and Cpu regions, respectively. This downregulated NE level might be due to the reduced noradrenergic activity caused by the oxidative damage of neurons in the brain of AD mouse[52]. Therefore, we further incubated the AD mouse brain slices with an antioxidant drug N-acetylcysteine (NAC). As shown in Fig. 5f, after NAC treatment, the fluorescence intensities of all the four regions were decreased as compared to AD samples, indicating the elevated concentrations of NE in the AD brain slices after NAC treatment, which returned to 87.2−96.8% of normal levels, and among these four regions, the S1BF region had the most significant increase in ratio (Fig. 5g). These results suggested a potential effect of the antioxidant NAC in improving the noradrenergic activity of neurons, which might be beneficial in relieving AD pathology.

## Discussion

In summary, we have developed a novel two-photon excited small molecular fluorescent probe BPS3 that enables real-time monitoring and imaging of norepinephrine at a single live-neuron level with highly selectivity and sensitivity, originating from a specific sequential nucleophilic substitution-cyclization trigger reaction. What's more, this developed probe can precisely target the neuronal cytomembrane, which is the frontier organelle at synapses that are directly involved in neurotransmitter secretion and uptake, playing an extremely important role in the transmission of nerve signals. The most notable feature of this probe is the impressively high temporal resolution that can response to NE within 100 ms, which is the fastest detection rate reported so far among small molecular fluorescent probes for NE. Based on the comprehensive experimental and theoretical analysis, we disclosed a unique dual acceleration mechanism, i.e., molecular conformational-folding and water-bridging, which significantly reduced the activation energy of the reaction between the probe and NE, thus increasing the reaction rate by more than $10^5$ and $10^3$ times, respectively. Benefiting from the excellent spatiotemporal resolution of the probe, we successfully applied it in imaging of

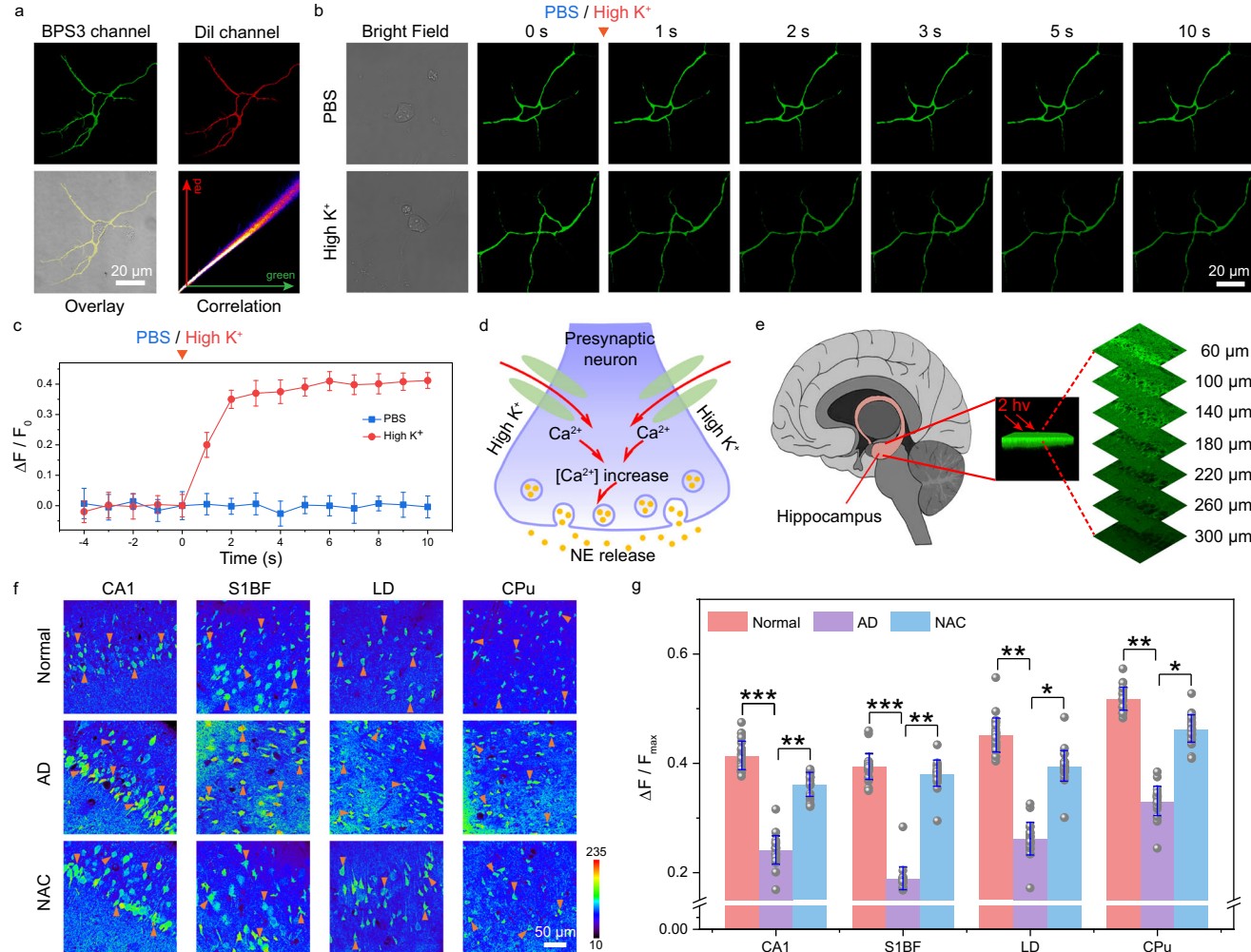

**Fig. 5 | Bioimaging performances. a** Confocal fluorescence images of neurons co-stained with BPS3 and a commercial membrane probe (DiI). Three independent experiments were repeated and similar results were obtained. **b** Time-lapse confocal fluorescence images of BPS3-incubated neurons stimulated by PBS buffer or high concentration of potassium, respectively. **c** Time-course of the fluorescence variation rate ($\Delta F/F_0$) of neurons stimulated by PBS (Phosphate-Buffered Saline) or high concentration of potassium, respectively (interval of 1 s). Data are presented as mean ± S.D. Error bars: S.D., $n = 5$ cells. **d** Illustration of the possible procedure of NE release upon stimulation with high concentration of potassium. **e** Three-dimensional two-photon confocal fluorescence images of the hippocampus region in mouse brain labeled with the BPS3 probe, two-photon excited at 720 nm. **f** Two-photon fluorescence images of the BPS3-incubated tissue slices from cornu ammonis of hippocampus (CA1), primary somatosensory cortex (S1BF), later-odorsal thalamic nucleus (LD), and caudate putamen (CPu) regions of normal and Alzheimer's disease (AD) mouse brains, as well as the N-acetylcysteine (NAC)-treated AD mouse brain. Orange arrows point to the representative single cells. **g** Histogram of the relative fluorescence variation rates corresponding to panel (f). Statistical differences were analyzed by Student's one-sided $t$-test. Data are presented as mean ± S.D. Error bars: S.D., $n = 25$ cells. $p = 0.0006$, 0.0083, 0.0007, 0.0074, 0.0077, 0.0156, 0.0068, 0.0233 from left to right, respectively, $*p < 0.05$, $**p < 0.01$, and $***p < 0.001$.

neuronal NE response and its dynamics under high potassium stimulation, which for the first time snapshotted the fast elevation of NE concentration at neuronal cytomembranes on a seconds-level time-scale by a small molecular fluorescence probe. What's more, an imaging study of brain tissue slices further exhibited a downregulated NE level in AD mouse, which might be due to the impaired noradrenergic function of neurons under AD pathology. The results presented here not only provided a superior molecular imaging tool for NE detection that is beneficial for advanced neuroscience research, but also revealed a dual acceleration mechanism that may further bring new inspiration to biological, analytical, organic, and physical chemists in the study of ultrafast detections and reaction kinetics regulations.

## Methods
### Materials
Unless stated specifically, all chemicals and reagents were purchased from commercial suppliers and were used without further purification.

Analytical grade solvents and all reagents were purchased from Sinopharm Chemical Reagent Co. Ltd. (Shanghai, China). Neurobasal medium, trypsin and B27 were purchased from Thermo Fisher Scientific. (U.S.A.). Column chromatography was carried out using silica gel.

### Syntheses
The synthetic procedures and details of the compounds mentioned in this report can be found in the Supplementary Information file.

### Instruments and kinetics study procedures
[1]H NMR, [13]C NMR spectra were measured on a Bruker 500 MHz spectrometer, and 2D NOESY/ROESY NMR spectra were measured on a Bruker 600 MHz spectrometer (Bruker, Germany). High-resolution mass spectrometry was performed on a Bruker ESI time-of-flight mass spectrometer and a Thermo Scientific Q Exactive Instrument (Germany). The absorption and fluorescence spectra were recorded in a rectangular quartz cell (10 × 10 × 45 mm) on the Hitachi UH-5300

spectrometer (Japan) and Hitachi F-4600 fluorescence spectrometer (Japan), respectively. The fast fluorescence response of the probes toward NE was carried out by using a stopped-flow accessory with a pneumatic drive system (SFA-20, HI-Tech, TgK Scientific, United Kingdom).Leica TCS-SP8 fluorescence microscope equipped with an oil immersion objective (40×) was used for fluorescence confocal imaging. The apoptosis assay was recorded on a FACSCalibur flow cytometry (Becton, Dickinson and Company, USA). All tests were performed at 298 K unless specifically mentioned.

The fast fluorescence response of the probes toward NE was carried out by using a stopped-flow accessory with a pneumatic drive system (SFA-20, HI-Tech, TgK Scientific, United Kingdom), combined with a fluorescence spectrometer (Hitachi F-4600, Japan). Equal volumes of the probe solution (10 μM, 150 μL, syringe A) and pure water (150 μL, syringe B) were rapidly driven from syringes into a highly efficient mixer and the basal fluorescence intensity of the probe was monitored at 480 nm. Subsequently, NE (200 nM, 150 μL, syringe B') and the probe (10 μM, 150 μL, syringe A) solutions were rapidly driven from both syringes to the mixer in the same way to initiate the fast reaction. The resultant reaction volume then displaced the contents of the optical cell (5 μM, 300 μL of the probe solution), thus filling it with freshly mixed reagents. During the entire mixing process, the fluorescence intensity was continuously monitored. All the injected volume was limited by a stop syringe that provided the "stopped-flow". The dead time of such mixing system was ca. 8 ms, and the fluorometer has a shorter sampling interval (5 ms).

### Theoretical and statistical calculations

Theoretical calculations have been performed using the DFT method implemented in the Gaussian 16 program package (Revision C.01)[53]. Molecular geomtries of the model complexes were optimized applying the PBE0-D3BJ[42,44] functional with the SMD[43] solvation model and water as the solvent. The Stuttgart-Dresden Stuttgart-Dresden (SDD)[41] effective core potential (ECP) was used to describe Br. For all the other atoms the 6-31 G* Pople basis set was used. As soon as the convergences of optimizations were obtained, the frequency calculations ate the same level have been performed to identify all the stationary points as minima or transition states, which has the unique imaginary frequencies. The intrinsic reaction coordinate (IRC)[54] calculations have been carried out to confirm that the transition structures can indeed connect the related reactant and product. All of the optimized geometries mentioned were built by Gaussview 6.0. The independent gradient model based on Hirshfeld partition (IGMH) was completed by using the Multiwfn program (3.8 dev)[55].

The limit of detection (LOD) was calculated as follows: LOD = 3σ/S, where σ is the standard deviation (S.D.) of the measured fluorescence intensities of the probe solution at 480 nm:

$$\sigma = \sqrt{\frac{\sum_{i=1}^{n}(x_i - \bar{x})^2}{n-1}} \tag{1}$$

where, n is the measured sample number, $x_i$ is the fluorescence intensities of each measurement, $\bar{x}$ is the averaged value for $x_i$.

### Culture of mouse cortical neurons

All animal care and in vivo experiments were performed according to Animal Care and Use Committee of East China Normal University (Shanghai, China). Mice had free access to water and food, and were kept on a 12 h light-dark cycle. C57BL/6 wild-type (WT) mice were provided by the Animal Care and Use Committee of East China Normal University, Shanghai, China. Newborn within 24 h C57BL/6 wild-type mice were anesthetized with halothane, and then the whole brain tissues were removed quickly and put in Hanks' balanced salt solution

(HBSS, free of $Mg^{2+}$ and $Ca^{2+}$) in an ice bath. Mouse cortical tissues were quickly striped and cultured in papain for 15 min at 37 °C, after that they were dispersed into poly-d-lysine-coated 35 mm Petri dishes at a density of $1 \times 10^6$ cells/dish. Neurons were cultured with neurobasal medium containing L-Glutamine and B27 (37 °C, 5% $CO_2$, 95% $O_2$) and the medium was changed three times a week.

### Apoptosis assay and cytotoxicity analysis

For the apoptosis assay, preincubated neurons were cultured with different concentrations of the probe BPS3 (0, 5, 10, 15, 20, 25, 30 μM) for 24 h. After removing the culture media, the cells were collected with the help of EDTA-free trypsin. After washing with PBS (1 mL × 3 times), the cells were re-suspended in 300 μL of binding buffer and further incubated by FITC-Annexin V and PI to label the apoptosis cells and necrotic cells, respectively. The flow cytometry data was analyzed by FlowJo (X 10.0.7 R2). For the cytotoxicity analysis, different concentrations of the probe BPS3 (0, 5, 10, 15, 20, 25, and 30 μM) were added into preincubated neurons in 96-well plates and cultured for 24 h. Subsequently, 20 μL of MTT was added to each well. The mixed solution was removed after reacting for 4 h, and 80 μL of DMSO was added. After shaking for 5 min, the absorbance at 490 nm was measured, and the cell viabilities were calculated according to the formula: cell viability (%) = absorbance of the experimental group/absorbance of the blank control group × 100%.

### Preparation and imaging of mouse brain tissue slices

All animal care and in vivo experiments were performed under the permission of Animal Care and Use Committee of East China Normal University (Shanghai, China). Due to the small sample size, sex of the mouse was not considered in the study design, and information on sex of mouse was not collected in the study. Fresh mouse brain tissue sections were prepared from 5-month-old normal mice (C57BL/6) and AD mice (APP/PS1), which were purchased from the Laboratory Animal Center of the Chinese Academy of Science. AD mice in the drug treatment group were intraperitoneally injected with N-acetylcysteine (NAC) (10 mg/kg/day) for 20 consecutive days, and mice in the control groups were intraperitoneally injected with the same amount and frequency of normal saline. First, the German Leica VT3000 vibrating blade microtome with a thickness of about 400 μm was used to obtain fresh mouse brain tissue slices, which was fully operated in ice-cold artificial cerebrospinal fluid (ACSF, NaCl 124.0 mM, KCl 3.0 mM, NaHCO3 26.0 mM, NaH2PO4 1.24 mM, MgSO4 8.0 mM, CaCl2 0.1 mM and D-glucose 10.0 mM) under a 95% $O_2$ and 5% $CO_2$ atmosphere. Next, the sections were transferred to ACSF containing 20.0 μM BPS3 and incubated at 37 °C for 60 min. ACSF was filled with 95% $O_2$ and 5% $CO_2$. The processed sections were washed with ACSF at least three times for imaging. Finally, a TCS-SP8 confocal laser scanning microscope equipped with a multiphoton laser was used to observe the stained sections.

### Reporting summary

Further information on research design is available in the Nature Portfolio Reporting Summary linked to this article.

## Data availability

All data supporting the findings of this study are available in this paper and the Supplementary Information.

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

## Acknowledgements

This work was supported by the National Natural Science Foundation of China (NSFC, 22274055 and 21904040 to Q.-W.Z.; 21635003, 21827814, and 21811540027 to Y.T.), National Key Research and Development Program of China (2022YFF0710000 to Y.T.), and the Fundamental Research Funds for the Central Universities (to Y.T.).

## Author contributions

L.M., Q.-W.Z., and Y.T. conceived and designed the experiments. L.M. performed the experiments. L.M., Y.H, Q.-W.Z. and Y.T. analyzed the data. L.M., Q.-W.Z., and Y.T. co-wrote the paper.

## Competing interests

The authors declare no competing interests.
