## [Peer Review File · Nature Communications]

Two-photon fluorescence imaging and specifically biosensing of norepinephrine on a 100-ms timescaleReviewers' Comments:

Reviewer #1:

Remarks to the Author:

The work by Mao et al. reported a new fluorescence probe to measure norepinephrine. Measuring NE is a significant area of research. The proposed new dyes seem promising. The following concerns need to be addressed, especially various experimental procedures are lacking, before final decision can be made.

1. Fluorescence spectra corresponding to Fig 1C on selectivity should be provided, with similar format as Fig. 1 a.
2. Introduction, the authors stated "capture of neurotransmitters by synapses..."⁵. Check the accuracy of this statement. While the authors cited ref 5, after looking into ref 5, this statement was not found.
3. One critical parameter is limit of detection, the authors used "population standard deviation" to calculate, it is not clear how did the author calculate the population standard deviation. The related experimental procedure should be included. In statistics, there is a way to estimate population standard deviation from the sample standard deviation.
4. Experimental procedure for data on Fig. 2a (fluorescence response dynamics) should be provided. It is not clear what instrument is used, and how the data is generated. How was the 100 ms determined? this information is closely related to the temporal resolution. Would the time scale between decrease and reach plateau depend on the concentration of NE?
5. Page 16: the authors stated that "excellent ability in neuronal cytomembrane targeting, which may be derived from the strong affinity between the positively charged pyridiniums in BPS3 and the negatively charged cell membrane". It is not clear which part of the cell membrane is BPS3 attached to? How would this affect the spatial resolution in NE study?
6. Figure 1a: Which functional group is the fluorescence generated from? Why adding NE affect this?
7. Discussion on Fig 4F should be expanded. As it is now, the discussion is rather brief, making it hard to follow, the readers are not sure what to look for in this figure.
8. Typo, abstract, "response" should be "respond"
9. Experimental procedure on measuring reaction kinetics should be provided. "It was boosted by >105 times by the new probe", why?
10. he presented studies involved incubation of the developed dyes with neurons, thus is not non-invasive. The statement the authors mentioned in the introduction and other places about "non-invasively monitor biomolecules" is misleading.

Reviewer #2 (Remarks to the Author):

see attached file

Norepinephrine (NE) is a very important neurotransmitter, and the specific and rapid detection of NE is crucial to understand the physiological and pathological functions of NE in living systems. In this work, Tian and co-workers developed a simple yet efficient small molecular fluorescent probe that could precisely anchor to the neuronal membrane and specifically respond to NE on a 100-ms-timescale. More importantly, they proposed a unique dual-acceleration mechanism, and disclosed the important role of the molecular-folding and the transition state of a water-bridged six-membered ring, which is a quite interesting finding. This is the first paper to systematically study the kinetics of NE detection, and provides a small-molecular tool to capture transient NE fluctuations, which will facilitate the research of complex neurotransmission with high spatiotemporal resolution. Therefore, I recommend the urgent publication in Nature Communications.

1. Considering that the probe molecule consists of two non-conjugated aromatic moieties, so which group does the fluorescence emission come from?
2. In order to confirm the influence of molecular folding on reaction kinetics, the authors designed and synthesized a series of control compounds. Here I suggest whether it is possible to prepare a "half-molecule" probe containing the reaction site, to see how the reaction rate is. I think this may further support the author's inference.

Reviewer #3:

Remarks to the Author:

I have reviewed the manuscript (Manuscript Number: NCOMMS-22-50884) entitled "Two-Photon Fluorescence Imaging and Specifically Biosensing of Norepinephrine on a 100-ms Timescale" submitted to Nature Communications. The authors developed a novel probe (BPS3), which selectively reacted with norepinephrine (NE) on 100-ms timescale. The mechanism of the fast and selective reaction was carefully investigated by spectroscopic measurements and density functional theory calculation. The authors also demonstrated the applicability of the probe to monitoring NE release in living neurons and mouse brains. The data and results are interesting. However, some deficiencies in experimental explanations exist in the current manuscript.

1) In Fig. 4b, the authors evaluated the response of BPS3 by monitoring DiI signals. The authors should explain why they did not monitor BPS3 directly.

2) The authors should describe the supplier of Alzheimer's disease mice and procedures of N-acetylcysteine treatment.

3) In Fig. 4g, the bar graph should include statistical significance (e.g., Student's t-test).

4) In section 1.4 of SI, the authors should add references regarding neuron cultivation.

5) The authors used the independent gradient model based on Hirshfeld partition (IGMH) to check the stability of the folded conformation of BPS3. However, the only software which can operate IGMH is Multiwfn. The authors should add the reference regarding Multiwfn and mention that the IGMH was completed by Multiwfn in Supplementary Information 1.3.

6) In Figure S35 (a), the peak derived from 5,7 seems to be one peak. However, in S35 (b) and (c), the peak from 5' and 7' split into two peaks. The authors should explain why this phenomenon appeared.

Responses to the reviewers' comments

Reviewer #1:

The work by Mao et al. reported a new fluorescence probe to measure norepinephrine. Measuring NE is a significant area of research. The proposed new dyes seem promising. The following concerns need to be addressed.

Comment 1: Fluorescence spectra corresponding to Fig. 1c on selectivity should be provided, with similar format as Fig. 1a.

Response: We appreciate the reviewer's high evaluation, positive recommendation, as well as the valuable comments.

As suggested by the reviewer, we have now supplemented the original fluorescence response plots with regard to the selective experiments. As shown in Figure R1a, except for NE, the addition of other related neurotransmitters hardly affected the fluorescence emission of the probe. Then, each fluorescence variation rate caused by different analytes were calculated ($(F_0 - F_i)/F_0$, where F_0 referred to the original fluorescence intensity of the probe, and F_i referred to the fluorescence intensity of the probe upon addition of each analyte), and normalized to the value of NE response (referred to 100%). The resulting relative selectivity diagram was shown in Figure R1b (same as Fig. 1c in the main text), which verified the superb selectivity of probe **BPS3** toward NE.

Figure R1a has been added and highlighted as Figure S37, on Page 23 in the revised Supplementary Information.

Figure R1. (a) Fluorescence spectra of **BPS3** (5 μ M) upon addition of different analytes, measured after ten minutes of mixing. The concentrations of NE, EP, and DA were 300 nM, and other species were all 1 mM. (b) Selectivity diagram of **BPS3** to different analytes corresponding to panel (a).

Comment 2: In introduction, the authors stated “capture of neurotransmitters by synapses...”⁵. Check the accuracy of this statement. While the authors cited ref 5, after looking into ref 5, this statement was not found.

Response: Thanks for the valuable suggestion. We have found a more comprehensive and precise literature and revised the statement about neuronal signaling between synapses. According to Ref. R1 (*Nature* **2014**, 515, 293-297.) and the schematic diagram below (Figure R2, from Ref. R1), although there are four hypotheses to describe the vesicle fusion and the subsequent neurotransmitter release process in the presynaptic neuron, namely, (i) Kiss and run, (ii) Full-collapse fusion, (iii) Ultra-fast endocytosis, and (iv) Bulk endocytosis, the general process of chemical signaling between neurons is widely recognized, i.e., when an electrical pulse reaches the tip of the neuronal axon, the vesicles there respond by moving to the synaptic membrane, merging with it and releasing the neurotransmitters, which then migrate to the neighboring neuron across the synaptic cleft, where they are bound to the receptors located in the postsynaptic neuronal membrane and initiate neuronal responses.

Therefore, we have revised the statement as: “ Signaling between neurons occurs when neurotransmitters are released from the presynaptic membrane, diffuse across the synapse to neighboring neurons, and bind to receptors on the postsynaptic neuronal membrane.”

This new statement has been highlighted in Line 3 to 6 on Page 3 in the revised manuscript, and Ref. R1 was added as the new Ref. 5 in the revised manuscript.

Figure R2. Schematic of the neurotransmitter-based signaling between neurons. (from Ref. R1)

Reference R1: Marx, V. A deep look at synaptic dynamics. *Nature* **2014**, 515, 293-297.

Comment 3: One critical parameter is limit of detection, the authors used “population standard deviation” to calculate, it is not clear how did the author calculate the population standard deviation. In statistics, there is a way to estimate population standard deviation from the sample standard deviation.

Response: Thanks very much for the comment. We actually used the standard deviation (σ), which was written as the population deviation by mistake. In general, the limit of detection (LOD) was calculated as follows: $\text{LOD} = 3\sigma / S$, where σ is the standard deviation of the measured fluorescence intensities of the probe solution at 480 nm:

$$\sigma = \sqrt{\frac{\sum_{i=1}^n (x_i - \bar{x})^2}{n - 1}}$$

where, $n = 20$, x_i is the fluorescence intensities of the probe of each measurement, \bar{x} is the averaged value for x_i , thus σ is calculated to be 0.00055.

S is the slope of calibration curve (0.0033), so the LOD was calculated:

$$\text{LOD} = 3\sigma / S = 3 \times 0.00055 / 0.0033 = 0.5 \text{ nM.}$$

We have corrected the statement and highlighted in Line 2-3 from the bottom on Page 6, in the revised manuscript. The calculation details were added and highlighted in the second paragraph on Page 3 in the revised Supplementary Information.

Comment 4: Experimental procedure for data on Fig. 2a (fluorescence response dynamics) should be provided. It is not clear what instrument is used, and how the data is generated. How was the 100 ms determined? Would the time scale between decrease and reach plateau depend on the concentration of NE?

Response: Thanks for the suggestion. The detailed procedures for the fluorescent kinetics studies are presented as follows:

The fast fluorescence response of the probes toward NE was carried out by using a stopped-flow accessory with a pneumatic drive system (Figure R3a) (SFA-20, HI-Tech, TgK Scientific, United Kingdom), combined with a fluorescence spectrometer (Hitachi F-4600, Japan). As schemed in Figure R3b, equal volumes of the probe solution (10 μM , 150 μL , syringe A) and pure water (150 μL , syringe B) were rapidly driven from syringes into a highly efficient mixer and the basal fluorescence intensity of the probe was monitored at 480 nm. Subsequently, NE (200 nM, 150 μL , syringe B') and the probe (10 μM , 150 μL , syringe A) solutions were rapidly driven from both syringes to the mixer in the same way to initiate the fast reaction. The resultant reaction volume then displaced the contents of the optical cell (5 μM , 300 μL of the probe solution), thus filling it with freshly mixed reagents. During the entire mixing process, the fluorescence intensity was continuously monitored. All the injected volume was

limited by a stop syringe that provided the “stopped-flow”. The dead time of such mixing system was ca. 8 ms, and the fluorometer has a shorter sampling interval (5 ms). So, in principle, any dynamic processes longer than this timescale can be monitored by this instrument, ensuring sufficient temporal resolution to measure the fluorescent response of the developed probes toward NE.

We have added and highlighted these experimental procedures to Section 1.2, on Page 2 in the revised Supplementary Information, and the instrument was also added and highlighted in Lines 6-8, on Page 23 in the revised manuscript.

Figure R3. (a) Picture of the stopped-flow accessory with a pneumatic drive system. (b) Schematic of the stopped-flow-based kinetics measurement.

Regarding the second question, we have measured the response time of the probe to different concentrations of NE (50, 100, 200, 300 nM). As shown in Figure R4, different concentrations of NE required different time to equilibrate the reaction (40–200 ms), where the lower the concentration, the shorter the time required. As reported, the physiological concentration of NE in neurons was in the range of 1–100 nM. Therefore, we chose to use this concentration uniformly in the main studies to evaluate and compare the response kinetics of different probes under various conditions. As shown in the Figure R4, by examining the fluorescence response curve, starting from the point where the intensity began to decline, and ending at the point where the intensity dropped to a stable plateau, and calculating the time interval between the two points, we could see that at the concentration of 100 nM of NE, the response time was about 100 ms. If the concentration was lower, its response time would be shorter. Therefore, this result have verified that the developed probe **BPS3** could respond to NE within 100 ms at physiological concentrations of NE.

We have added highlighted this discussion in Lines 7-10, on Page 11, in the revised manuscript, and added Figure R4 as Figure S47 on Page 27 in the revised Supplementary Information.

Figure R4. Normalized fluorescence response dynamics (recorded at 480 nm) of 5 μ M aqueous solution of **BPS3** with addition of various concentrations of NE (50, 100, 200, and 300 nM).

Comment 5: Page 16: the authors stated that “excellent ability in neuronal cytomembrane targeting, which may be derived from the strong affinity between the positively charged pyridiniums in **BPS3** and the negatively charged cell membrane”. It is not clear which part of the cell membrane is **BPS3** attached to? How would this affect the spatial resolution in NE study?

Response: Although our confocal microscope could not provide higher resolution to see where exactly the probe was localized in the cell membrane, a high Pearson’s coefficient value of 0.93 was obtained through the co-localization experiment with the commercial cell membrane tracer DiI (Figure R5a), indicating that they had very similar targeting capabilities. Given their similar amphiphilic structures, with positively charged heads and lipophilic tails (Figure R5b), it is understandable that they have similar targeting capabilities that can uniformly label the entire neuron membrane.^{R2} On the other hand, by overlaying the fluorescent and bright field images of neurons, we could also observe that the dye was evenly distributed on the membrane of neurons. Other probes with similar pyridinium structures have also been reported to have cell membrane labeling ability.^{R3,R4} It is proposed that the cationic pyridinium have electrostatic affinity with the phosphate anion of the cell membrane surface, while the hydrophobic phenyl groups can be embedded into the cell membrane (Figure R5c). Therefore, the current experiments have clearly demonstrated that the probe possesses general membrane labeling ability, while it is not yet possible to tell whether it has a more precise submembrane spatial resolution.

We have added highlighted this discussion in Lines 3-6 from the bottom, on Page 18 in the revised manuscript, and added Figure R5c as Figure S56, on Page 32 in the revised Supplementary Information. Ref. R2–R4 were cited as Ref. 46, Ref. 47, and Ref. 48, respectively, in the revised manuscript.

Figure R5. (a) Confocal fluorescence images of neurons co-stained with **BPS3** and a commercial membrane probe (DiI). (b) Chemical structures of DiI and **BPS3**. (c) Schematic representation of **BPS3** targeting to cell membranes.

Reference R2: Beattie, E. C.; Howe, C. L.; Wilde, A.; Brodsky, F. M.; Mobley, W. C. NGF Signals through TrkA to Increase Clathrin at the Plasma Membrane and Enhance Clathrin-Mediated Membrane Trafficking. *J. Neurosci.* **2000**, *20*, 7325-7333.

Reference R3: Niu, N.; Yu, Y.; Zhang, Z.; Kang, M.; Wang, L.; Zhao, Z.; Wang, D.; Tang, B. Z. A cell membrane-targeting AIE photosensitizer as a necroptosis inducer for boosting cancer theranostics. *Chem. Sci.* **2022**, *13*, 5929-5937.

Reference R4: Liu, Y.; Zhang, T.; Huo, F.; Yin, C. Pyridine salts and aliphatic chains regulating membrane-targeted ratiometric fluorescence probe for detection of SO₂ in living cells. *Dyes Pigments* **2022**, *205*, 110540.

Comment 6: Figure 1a: Which functional group is the fluorescence generated from? Why adding NE affect this?

Response: In order to understand the source of the fluorescence and its variation, we further synthesized two reference compounds, namely **R1**, **R2** (Figure R6a), which represented the two fragment moieties of the probe **BPS3**, respectively. As shown in Figure R6b, the *p*-bromophenyl

pyridinium moiety (**R1**) of the probe emitted little fluorescence, while the *S*-phenyl carbonothioate-containing pyridinium moiety (**R2**) exhibited similar fluorescence emission to the probe **BPS3**. Therefore, it could be inferred that the fluorescence of the probe was generated from the *S*-phenyl carbonothioate-containing pyridinium moiety. Next, in order to rationalize the fluorescence response caused by the cleavage reaction triggered by NE, we further synthesized the third reference compound (**R3**) representing the fragment moiety of the product **BPS3-OH** (Figure R6a). It could be seen from Figure R6b that after cleavage of the *S*-phenyl carbonothioate group, and form the hydroxyl product **R3**, the fluorescence was significantly quenched as compared with **R2**. We deduced that in both the reference compound **R3** and product **BPS3-OH**, the hydroxyl group acted as an electron-donating group and the pyridinium acted as the electron-withdrawing group, thus generating a twisted intramolecular charge transfer (TICT) state, which greatly enhanced the non-radiative decay thus quenched the fluorescence emission. To verify this inference, the viscosity-dependent absorption spectra were measured for **BPS3-OH**. As shown in Figure R7a, as the viscosity of the solvent increased by increasing the fraction of glycerol in solution, the absorption maximum underwent an obvious red-shifting from 354 nm to 362 nm. Such viscosity-dependent solvatochromism clearly confirmed the TICT feature of the compound **BPS3-OH**. Moreover, to get a deep insight into the photophysical property of **BPS3-OH**, the femtosecond transient absorption spectrum was measured in PBS (10 mM, pH = 7.4) with the probe delay time from 0.001 ps to 200 ps. As shown in Figure R7b, the initial evolution spectrum of **BPS3-OH** from 0.1 ps to 2.2 ps showed a significant negative signal from 400 nm to 600 nm, which was consistent with fluorescence spectrum and could be assigned to the stimulated emission (SE) of locally excited (LE) state. Next, this negative signal underwent decay accompanied by the appearance of a positive signal from 590 nm to 720 nm which was attributed to the absorption of LE state. At about 4.6 ps, another absorption signal at 460 nm appeared which slightly blue-shifted with time evolution, and vanished at around 30 ps. This signal was assigned to the absorption of TICT state.^{R5} Therefore, all these results have confirmed the TICT characteristics of the NE-triggered cleaved product **BPS3-OH**. On the contrary, the TICT process did not occur in the probe **BPS3** due to the existence of electron-withdrawing groups on both sides of the fluorophore. This may be the reason for the significantly quenched fluorescence emission of the probe after reaction with NE.

We have added and highlighted this discussion in the second paragraph on Page 9 and Page 10 in the revised manuscript. We also added the synthesis procedures and characterizations of the newly synthesized reference compounds **R2**, **R3** on Pages 19-21, and added Figure R6 as Figure S44, and Figure R7 as Figure S45 on Page 26, in the revised Supplementary Information. Ref. R5 was cited as Ref. 28 in the revised manuscript.

Figure R6. (a) Chemical structures of the referred compounds. (b) Fluorescence spectra of the corresponding compounds in panel (a), 10 μM in PBS (10 mM, pH=7.4), excited at 360 nm.

Figure R7. (a) Viscosity-dependent normalized absorption spectra of **BPS3-OH** (10 μM). (b) Transient absorption spectrum of **BPS3-OH**, 1 mM in PBS (10 mM, pH=7.4). The color bar showed the relative optical density.

Reference R5: Li, Y.; Liu, X.; Han, J.; Cao, B.; Sun, C.; Diao, L.; Yin, H.; Shi, Y. Solvent viscosity induces twisted intramolecular charge transfer state lifetime tunable of Thioflavin-T. *Spectrochim. Acta A Mol. Biomol. Spectrosc.* **2019**, *222*, 117244.

Comment 7: Discussion on Fig 4F should be expanded. As it is now, the discussion is rather brief, making it hard to follow.

Response: Thanks for the suggestion, we have conducted a more detailed discussion on the imaging of brain tissue slices, as follows:

We prepared acute brain tissue slices from four different regions of both AD and normal mice, namely, cornu ammonis of hippocampus (CA1), primary somatosensory cortex (S1BF), laterodorsal thalamic nucleus (LD), and caudate putamen (CPu), and then cultured with the probe **BPS3**. Confocal microscopic images of these tissue slices were presented in Figure R8a (same as Fig 4f in the

manuscript), where the pseudocolor ranging from blue to red represented increase of the relative fluorescence intensity. Then, by randomly selecting 25 cells in each image, followed by statistical analysis on their fluorescence intensity variations, the histogram was generated as Figure R8b. The statistical results indicated some inhomogeneity of NE distribution in various brain regions. It is worth noting that under the AD model, the NE content in all these brain regions decreased significantly to 58.3%, 48.0%, 58.0%, and 63.9% of the normal level, in CA1, S1BF, LD, and Cpu regions, respectively. This downregulated NE level might be due to the reduced noradrenergic activity caused by the oxidative damage of neurons in the brain of AD mouse. Thereafter, we further incubated the AD mouse brain slices with an antioxidant drug N-acetylcysteine (NAC). As shown in Figures R8a, after NAC treatment, the fluorescence intensities of all the four regions were decreased as compared to AD samples, indicating the elevated concentrations of NE in the AD brain slices after NAC treatment, which returned to 87.2–96.8% of normal levels, and among these four regions, the S1BF region had the most significant increase in ratio (Figure R8b). These results suggested a potential effect of the antioxidant NAC in improving the noradrenergic activity of neurons, which might be beneficial in relieving AD pathology.

We have added and highlighted this discussion in Lines 3-17, on Page 21, in the revised manuscript.

Figure R8. (a) Two-photon fluorescence images of the **BPS3**-incubated tissue slices from CA1, S1BF, LD, and CPu regions of normal and AD mouse brains, as well as the NAC-treated AD mouse brain. (b) Histogram of the relative fluorescence variation rates corresponding to panel (a). Error bars: S.D., $n = 25$, * $p < 0.05$, ** $p < 0.01$, and *** $p < 0.001$ (Student's t test).

Comment 8: Typo, abstract, “response” should be “respond”

Response: Thanks for the correction. We have corrected this typo and highlighted in Line 5 on Page 2 in the revised manuscript.

Comment 9: Experimental procedure on measuring reaction kinetics should be provided. “It was boosted by $>10^5$ times by the new probe”, why?

Response: For the detailed experimental procedures of the kinetics studies, please see the response to **Comment 4**. Regarding the acceleration rate ($>10^5$) of the reaction rate induced by molecular conformational folding, we compared the reaction kinetics of the probe **BPS3** and the control compound **BPS2**, respectively. As confirmed in the main text, probe **BPS3** possessed a folded conformation, while for the control compound **BPS2**, there were only two carbons in the alkyl chain between the two rigid phenyl groups, thus it could hardly form a folded conformation due to the larger molecular bond tension and steric hindrance (Figure R9a). Whereafter, the reaction kinetics of these two compounds with NE were subsequently studied. It was found that despite with only one-carbon shorter in the alkyl chain, compound **BPS2** hardly produced any response to NE at room temperature, even extending the reaction time to 7 days (Figure R9b). Even if the temperature was elevated to 50 °C, it still could not achieve the reaction equilibrium within 12 hours (Figure R9c). In order to get a better kinetic fitting curve, we continued to increase the temperature to 60 °C, which could reach the reaction equilibrium within 8 hours (Figure R9d), so that we were able to calculate a k_{obs} value of 0.5818 h^{-1} (equal to $1.6 \times 10^{-4} \text{ s}^{-1}$) for the reaction between **BPS2** and NE at 60 °C (Figure R9e). In contrast, the probe **BPS3** that possessed a highly similar structure but with folded conformation, could complete the reaction to NE within 100 ms (Figure R9f), thus the k_{obs} value was calculated to be 0.0532 ms^{-1} (equal to 53.2 s^{-1}) (Figure R9g). Therefore, by comparing the k_{obs} values for the two probes with NE, the acceleration rate could be calculated as: $53.2 / (1.6 \times 10^{-4}) = 3.325 \times 10^5$. Moreover, this acceleration rate was calculated by using the k_{obs} values of the reaction for **BPS2** at 60 °C (because no reaction was detected at room temperature, Figure R9b), while for **BPS3** at room temperature. Therefore, the actual acceleration rate at room temperature would be much higher than 10^5 times.

Figure R9. (a) Chemical structures of compounds **BPS2** and **BPS3**. Normalized fluorescence response dynamics of **BPS2** (5 μ M) reacted with NE (100 nM) at room temperature (b), 50 $^{\circ}$ C (c), and 60 $^{\circ}$ C (d), respectively. (e) Kinetic fluorescence analysis corresponding to panel (d). (f) Normalized fluorescence response dynamics of **BPS3** (5 μ M) reacted with NE (100 nM) at room temperature. (g) Kinetic fluorescence analysis corresponding to panel (f).

Comment 10: The presented studies involved incubation of the developed dyes with neurons, thus is not non-invasive. The statement the authors mentioned in the introduction about “non-invasively monitor biomolecules” is misleading.

Response: Yes, thanks for the comment and we agree with the reviewer that fluorescence imaging technology itself can be non-invasive, and this statement only makes sense for in vivo experiments. However, most currently reported fluorescent probes (including the probe in this paper) are applied at the level of cells or tissue slices.

Therefore, we have removed the statement of “non-invasive” and revised the statement as follows: “Fluorescent probes combined with fluorescence imaging technology may serve as a promising candidate to monitor biomolecules, including neurotransmitters, in real-time with high sensitivity and spatiotemporal resolution.” It was highlighted in Lines 7-9 from the bottom, on Page 3, in the revised manuscript.

Reviewer #2:

In this work, Tian and co-workers developed a simple yet efficient small molecular fluorescent probe that could precisely anchor to the neuronal membrane and specifically respond to NE on a 100-ms-timescale. More importantly, they proposed a unique dual-acceleration mechanism, and disclosed the important role of the molecular-folding and the transition state of a water-bridged six-membered ring, which is a quite interesting finding. This is the first paper to systematically study the kinetics of NE detection, and provides a small-molecular tool to capture transient NE fluctuations, which will facilitate the research of complex neurotransmission with high spatiotemporal resolution. Therefore, I recommend the urgent publication in Nature Communications.

Comment 1: Considering that the probe molecule consists of two non-conjugated aromatic moieties, so which group does the fluorescence emission come from?

Response: We very much appreciate the reviewer's high evaluation and positive recommendation on our manuscript.

To answer this question, we further synthesized two reference compounds, namely **R1**, **R2** (Figure R10a), which represented the two fragment moieties of the probe **BPS3**, respectively. As shown in Figure R10b, the *p*-bromophenyl pyridinium moiety (**R1**) of the probe emitted little fluorescence, while the *S*-phenyl carbonothioate-containing pyridinium moiety (**R2**) exhibited similar fluorescence emission to the probe **BPS3**. Therefore, it could be inferred that the fluorescence of the probe was mainly generated from the *S*-phenyl carbonothioate-containing pyridinium moiety of the probe.

We have added and highlighted this discussion in the second paragraph on Page 9 in the revised manuscript. We also added the synthesis procedures and characterizations of the newly synthesized reference compounds **R2**, **R3** on Pages 19-21, and added Figure R10 as Figure S44 on Page 26, in the revised Supplementary Information.

Figure R10. (a) Chemical structures of the referred compounds. (b) Fluorescence spectra of the corresponding compounds in panel (a), 10 μ M in PBS (10 mM, pH=7.4), excited at 360 nm.

Comment 2: In order to confirm the influence of molecular folding on reaction kinetics, the authors designed and synthesized a series of control compounds. Here I suggest whether it is possible to prepare a “half-molecule” probe containing the reaction site, to see how the reaction rate is. I think this may further support the author’s inference.

Response: Thanks very much for the suggestion. We have synthesized the control compound **R2** as the “half-probe” and studied its reaction kinetics toward NE and compared it with the probe **BPS3**. As shown in Figure R11a, although it had the same reaction site as **BPS3**, the reaction rate of the control compound **R2** toward NE was much lower than that of the probe **BPS3**. Under the same conditions, it took about 257 seconds to reach the reaction equilibrium, while only 100 ms for the probe **BPS3** (Figure R11b), a difference of ca. 2570 times. Therefore, this experiment again verified the important role of the unique folded conformation of **BPS3** in accelerating the reaction kinetics.

We have added and highlighted this discussion in Lines 10-15, on Page 14, in the revised manuscript. We have also added Figure R11a as Figure S50 on Page 29, in the revised Supplementary Information.

Figure R11. Chemical structures and normalized fluorescence response dynamics of 5 μM aqueous solution of **R2** (a), and **BPS3** (b) toward NE.

Reviewer #3:

The authors developed a novel probe (**BPS3**), which selectively reacted with norepinephrine (NE) on 100-ms timescale. The mechanism of the fast and selective reaction was carefully investigated by spectroscopic measurements and density functional theory calculation. The authors also demonstrated the applicability of the probe to monitoring NE release in living neurons and mouse brains. The data and results are interesting. However, some deficiencies in experimental explanations exist in the current manuscript.

Comment 1: In Fig. 4b, the authors evaluated the response of **BPS3** by monitoring DiI signals. The authors should explain why they did not monitor BPS3 directly.

Response: We very much appreciate the reviewer's high evaluation and positive recommendation on our manuscript, as well as the valuable comments that greatly helped us improve this manuscript.

Regarding the fluorescence signal in Figure 4b, in fact it was the fluorescence signal from the probe **BPS3**, but we used a red pseudocolor, which might mislead the readers as it looked like the DiI signal in Figure 4a. Therefore, as shown in Figure R12 below, we have modified the imaging color of the neurons to green color for ease of reading.

This modified Figure 4b was highlighted on Page 19, in the revised manuscript.

Figure R12. (a) Confocal fluorescence images of neurons co-stained with **BPS3** and a commercial membrane probe (DiI). (b) Time-lapse confocal fluorescence images of **BPS3**-incubated neurons stimulated by PBS buffer or high concentration of potassium, respectively.

Comment 2: The authors should describe the supplier of Alzheimer's disease mice and procedures of N-acetylcysteine treatment.

Response: AD mice (APP/PS1) were purchased from the Laboratory Animal Center of the Chinese Academy of Science, and used at the age of 6–7 months. AD mice in the drug treatment group were intraperitoneally injected with N-acetylcysteine (NAC) (10 mg/kg/day) for 20 consecutive days, and mice in the control groups were intraperitoneally injected with the same amount and frequency of normal saline.

This experimental procedures were added and highlighted in Lines 4-8 of Section 1.6 on page 4, in the revised Supporting Infromation.

Comment 3: In Fig. 4g, the bar graph should include statistical significance (e.g., Student's t-test).

Response: Thanks for the suggestion, and we have marked the statistical significance according to the Student's t test analysis. The revised Fig. 4g was as below (Figure R13), and highlighted on Page 19, in the revised manuscript.

Figure R13. Histogram of the relative fluorescence variation rates in **BPS3**-stained CA1, S1BF, LD, and CPu regions of normal and AD mouse brains, as well as the NAC-treated AD mouse brain. Error bars: S.D., n = 25, *p < 0.05, **p < 0.01, and ***p < 0.001 (Student's t test).

Comment 4: In section 1.4 of SI, the authors should add references regarding neuron cultivation.

Response: Thanks for the suggestion. We have added two references (R6 and R7) regarding the neuron cultivation in section 1.4 of SI (as Refs. S8-S9), which were highlighted in Line 3 of Section 1.4 on Page 3, and Ref S8-S9 were listed and highlighted on Page 33, in the revised Supplementary Information.

Reference R6. Chen, C.; Pan, Y.; Li, D.; Han, Y.; Zhang, Q.-W.; Tian, Y. An Intramolecular Charge Transfer–Förster Resonance Energy Transfer Integrated Unimolecular Platform for Two-Photon Ratiometric Fluorescence Sensing of Methionine Sulfoxide Reductases in Live-Neurons and Mouse Brain Tissues. *Anal. Chem.* **2022**, *94*, 6289-6296.

Reference R7. Gong, Z.; Liu, Z.; Zhang, Z.; Mei, Y.; Tian, Y. A Highly Stable Two-Photon Ratiometric Fluorescence Probe for Real-Time Biosensing and Imaging of Nitric Oxide in Brain Tissues and Larval Zebrafish. *CCS Chemistry* **2021**, *3*, 2201-2211.

Comment 5: The authors used the independent gradient model based on Hirshfeld partition (IGMH) to check the stability of the folded conformation of **BPS3**. However, the only software which can operate IGMH is Multiwfn. The authors should add the reference regarding Multiwfn and mention that the IGMH was completed by Multiwfn in Supplementary Information 1.3.

Response: Thanks for the suggestion. We have supplemented the description and added the reference about the IGMH calculation as follows:

The independent gradient model based on Hirshfeld partition (IGMH) was completed by using the Multiwfn program.^{R8}

Reference R8. Lu, T.; Chen, F. Multiwfn: A multifunctional wavefunction analyzer. *J. Comput. Chem.* **2012**, *33*, 580-592.

This statement was highlighted in Lines 4-5, on Page 3, and the reference R8 was add as Ref. S7 on Page 33, in the in the revised Supplementary Information.

Comment 6: In Figure S35 (a), the peak derived from 5,7 seems to be one peak. However, in S35 (b) and (c), the peak from 5' and 7' split into two peaks. The authors should explain why this phenomenon appeared.

Response: In the probe **BPS3** (Figure R14a), since the bromo and carbonothioate groups on the two sides of the molecule both had similar electron-withdrawing feature, thus the chemical environment of the corresponding aromatic and alkyl protons on both sides had a higher symmetry, resulting in similar chemical shifts, which could be observed not only for H₅ ~ H₇, but also for H₄ ~ H₈, as well as H₃ ~ H₉ proton pairs (Figure R14a). On the contrary, the hydroxyl group in the product **BPS3-OH** formed by cleavage of the carbonothioate group, had an electron-donating property, which was quite different from the electron-withdrawing bromo group, resulting in a higher asymmetry of chemical environment for the corresponding protons on the left and right sides of the molecule, so the chemical shifts of these three pairs of protons split (H_{5'} ~ H_{7'}, H_{4'} ~ H_{8'}, H_{3'} ~ H_{9'}, Figure R14b and c).

Figure R14. ^1H NMR spectra of **BPS3** (a), **BPS3** reacted with NE for 60 mins (b), and **BPS3-OH** (c) in $\text{DMSO-}d_6$, at 298 K.

Reviewers' Comments:

Reviewer #1:

Remarks to the Author:

After the authors made changes, the information now is more accurate especially for the description of signaling between neurons at the synapses. While the authors have addressed most of my feedback, the discussion around Figure 4f can be improved, as it is now, it is not very clear. The authors added the following discussion about Fig. 4f, "Then, by randomly selecting 25 cells in each image, followed by statistical analysis on their fluorescence intensity variations, the histogram was generated as Figure 4g." It is not clear in Figure 4f, where are the single cells, if the authors could label a couple, that would help the readers understand this figure.

Another major concern is Figure 4b. The authors described "As shown in the Figure 4b, initially, both the two groups of neurons exhibited strong and comparative fluorescence emission at the neuron cytomembranes. However, when one set was treated with PBS buffer only, while the other with high concentration of potassium, the fluorescence intensities of the two groups differed greatly, that is, the fluorescence intensity of the former almost kept constant, while that of the latter varied significantly within 2 seconds". The changes on fluorescence intensity from 1s to 10s is not very obvious for potassium-stimulated sample, why?

Reviewer #2:

Remarks to the Author:

authors addressed all issues, i would like its publication in NC

Reviewer #3:

Remarks to the Author:

The authors have adequately revised their manuscript according to my previous comments and suggestions. The quality of the manuscript has been improved after the revision. I do not have further criticism of the work.

Responses to the reviewer's comments:

Reviewer #1:

Comment 1: The authors added the following discussion about Fig. 4f, “Then, by randomly selecting 25 cells in each image, followed by statistical analysis on their fluorescence intensity variations, the histogram was generated as Figure 4g.” It is not clear in Figure 4f, where are the single cells, if the authors could label a couple, that would help the readers understand this figure.

Response: We appreciate the reviewer's positive recommendation and valuable comments. As suggested by the reviewer, we have now marked a couple of single cells for each image with orange arrowheads for easier understanding (Fig. R1).

This modified Figure was highlighted as Fig. 5f on Page 19, in the revised manuscript.

Fig. R1 Two-photon fluorescence images of the BPS3-incubated tissue slices from CA1, S1BF, LD, and CPu regions of normal and AD mouse brains, as well as the NAC-treated AD mouse brain. Orange arrows point to some representative single cells.

Comment 2: The authors described “As shown in the Figure 4b, initially, both the two groups of neurons exhibited strong and comparative fluorescence emission at the neuron cytomembranes. However, when one set was treated with PBS buffer only, while the other with high concentration of potassium, the fluorescence intensities of the two groups differed greatly, that is, the fluorescence intensity of the former almost kept constant, while that of the latter varied significantly within 2 seconds”. The changes on fluorescence intensity from 1s to 10s is not very obvious for potassium-stimulated sample, why?

Response: Thanks very much for the comment. As shown in the Figs. R2a, b (corresponding to Figs. 5b, c in the revised manuscript), when the neurons were stimulated by high potassium, the fluorescence

intensities were significantly attenuated in the first 2 seconds (0–2 s) compared with that without high potassium stimulation (PBS only). Thereafter, the fluorescence signals hardly changed any more (2–10 s). It indicated that neurons could respond to external stimuli of high potassium and reach a steady state in a quite short time (within 2 s). However, the total fluorescence variation rate ($\Delta F/F_0$) was only ~40%, and the concentration of norepinephrine was estimated to be ~100 nM from the in vitro working curve (Fig. R2c). Therefore, the overall fluorescence intensity changes might not be very obvious from the naked eye, but still significant enough to be read by the microscopic instrument (Fig. R2b).

Fig. R2 (a) Time-lapse confocal fluorescence images of BPS3-incubated neurons stimulated by PBS buffer or high concentration of potassium, respectively. (b) Time-course of the fluorescence variation rate ($\Delta F/F_0$) of neurons stimulated by PBS (Phosphate Buffered Saline) or high concentration of potassium, respectively (interval of 1 s). Data are presented as mean \pm S.D. Error bars: S.D., $n = 5$ cells. (c) Plot and linear fitting of the fluorescence variation rates ($\Delta F/F_0$) at 480 nm versus the concentration of NE (0–300 nM). Data are presented as mean \pm S.D. Error bars: S.D., $n = 3$ independent experiments.